# Overcoming Resistance to FLT3 Inhibitors in the Treatment of *FLT3*-Mutated AML

**DOI:** 10.3390/ijms21041537

**Published:** 2020-02-24

**Authors:** Stephen S.Y. Lam, Anskar Y.H. Leung

**Affiliations:** Division of Haematology, Department of Medicine, Li Ka Shing Faculty of Medicine, The University of Hong Kong, Hong Kong, China; stephen_lsy127@me.com

**Keywords:** AML, FLT3, drug resistance

## Abstract

Acute myeloid leukaemia (AML) carrying internal tandem duplication (ITD) of Fms-Like Tyrosine kinase 3 (*FLT3*) gene is associated with high risk of relapse and poor clinical outcome upon treatment with conventional chemotherapy. FLT3 inhibitors have been approved for the treatment of this AML subtype but leukaemia relapse remains to be a major cause of treatment failure. Mechanisms of drug resistance have been proposed, including evolution of resistant leukaemic clones; adaptive cellular mechanisms and a protective leukaemic microenvironment. These models have provided important leads that may inform design of clinical trials. Clinically, FLT3 inhibitors in combination with conventional chemotherapy as induction treatment for fit patients; with low-intensity treatment as salvage treatment or induction for unfit patients as well as maintenance treatment with FLT3 inhibitors post HSCT hold promise to improve survival in this AML subtype.

## 1. Introduction

Acute myeloid leukaemia (AML) is defined pathologically by an abnormal increase in blasts in blood and/or bone marrow (BM). It is a group of heterogeneous diseases with distinct driver events and pathogeneses that may occur at different stages of the haematopoietic hierarchy [1]. Subtypes of AML show different morphologies, immunophenotypes as well as cytogenetic, genetic and clinical features. Conventional chemotherapy, comprising induction and consolidation as well as allogeneic haematopoietic stem cell transplantation (HSCT) are the mainstays of treatment but only 30%–40% of patients can achieve long-term remission. The outcome of elderly patients ineligible for chemotherapy and HSCT is dismal.

Laboratory studies in leukaemia biology in the past few decades have led to identification of molecular targets and development of novel therapeutic strategies. In the past 3 years, eight therapeutic agents have been approved by the U.S. Food and Drug Administration (FDA) for the treatment of different AML subtypes in different clinical contexts (Table 1). In particular, two multi-kinase inhibitors were approved for the treatment of AML carrying gain-of-function mutations in Fms-like tyrosine kinase 3 (*FLT3*) gene: Midostaurin in combination with conventional induction and consolidation chemotherapy in newly diagnosed patients [2] and gilteritinib monotherapy for relapsed/refractory (R/R) *FLT3*-mutated patients [3]. Another specific FLT3 inhibitor quizartinib was also approved in Japan for the treatment of R/R patients [4]. However, disease relapse remains an important cause of treatment failure. This review focuses on the potential mechanisms of drug resistance in this AML subtype and strategies that may be exploited to overcome resistance.

## 2. Fms-Like Tyrosine Kinase 3 (FLT3)

Fms-like tyrosine kinase 3 (*FLT3*), first identified to be expressed in normal haematopoietic stem and progenitor cells, is one of the most frequently mutated genes in AML. Internal tandem duplication (ITD) is the commonest genetic abnormality and is associated with leucocytosis at diagnosis and high risk of relapse after conventional chemotherapy, particularly those with high ITD allelic ratio [5], large ITD size [6] and multiple ITD clones [7]. *FLT3*-ITD occurs particularly in AML with normal cytogenetics, where it occurs in up to 40% cases, and those with rare t(6;9) translocation involving *DEK*/*CAN* gene fusion, where it occurs in up to 70%–80% [8]. Missense mutations of tyrosine kinase domain (TKD) also occur, albeit infrequently at 5%–10% of AML. Their prognostic impact has remained unclear [9], which could be due to their low incidence of occurrence or modest biologic activities. Mechanistically, both ITD and TKD mutations result in constitutive activation of FLT3 signalling, hence the cellular proliferation, anti-apoptosis and differentiation block that are often seen in *FLT3* mutant AML cases [1,10].

Under physiological conditions, FLT3 protein is activated by its ligand (FLT3L) and the binding results in dimerization and conformational changes of FLT3 that expose the phosphorylation sites of its TKD. Subsequent auto-phosphorylation of FLT3 leads to binding of adaptor proteins such as SHP2, Grb2 and SRC family kinases, hence activation of downstream signalling kinases including MAPK/ERK, JAK/STAT and PI3K/AKT/mTOR [11]. While both ITD and TKD mutations result in constitutive activation of FLT3, via MAPK and PI3K pathways, there are significant differences in the activated downstream signalling pathways between them [12]. For instance, *FLT3*-TKD is associated with activation of SHP1 and SHP2 phosphatases that negatively regulate JAK signalling, whereas STAT5 activation via SRC binding is only seen in *FLT3*-ITD but not *FLT3*-TKD or *FLT3*-WT cells [13].

## 3. FLT3 Inhibitors

The pathogenetic roles of *FLT3*-ITD and TKD in AML and the inferior outcome of this AML subtype provide the basis for developing FLT3 inhibitors. Mechanistically FLT3 inhibitors can be categorised into 2 types. Type I inhibitors bind FLT3 in the active conformation near the activation loop or ATP binding site and are effective against both ITD and mutant TKD as exemplified by midostaurin, sunitinib, lestaurtinib, crenolanib and gilteritinib. Type II inhibitors bind FLT3 in the inactive state near the ATP binding domain, targeting FLT3-ITD but not mutant TKD. Examples include sorafenib, ponatinib and quizartinib. In order of their development, FLT3 inhibitors can be categorised into those of first and second generation. First generation FLT3 inhibitors refer to several multi-kinase inhibitors including lestaurtinib, sunitinib, sorafenib and midostaurin that have been evaluated since early 2000. As monotherapy, sorafenib has been widely used as salvage therapy for R/R *FLT3*-ITD AML with a rate of combined complete remission (CR) and CR with incomplete haematologic recovery (CRi) at about 16%–46%. Responses were typically transient with median duration of 1–3 months, and sorafenib was perceived at best as a bridging therapy to curative allogeneic HSCT. More recent data showed that sorafenib may be effective as maintenance therapy post HSCT, resulting in improved survival of these patients [14,15,16]. Midostaurin showed only modest effect as monotherapy in R/R AML [17]. When used as an adjunct to conventional chemotherapy in newly diagnosed *FLT3*-mutated AML, midostaurin was shown to prolong overall survival [2], leading to its approval by FDA.

Second generation FLT3 inhibitors refer to new inhibitors with higher specificity and potency against FLT3, including quizartinib, gilteritinib and crenolanib. They have been tested in clinical trials as monotherapy in R/R *FLT3*-mutated AML with CR/CRi of 23%–57% and median duration of response of 9–20 weeks (Figure 1). Based on an interim analysis of the ADMIRAL trial [3], gilteritinib was granted FDA approval for the treatment of R/R *FLT3*-mutated AML. In a randomised open-label phase 3 study (QUANTUM-R) [4], quizartinib was shown to improve overall survival compared with salvage chemotherapy in R/R *FLT3*-mutated AML. Several phase 3 studies were underway including those that evaluate gilteritinib following induction and consolidation chemotherapy (NCT02236013, NCT02310321) as well as allogeneic HSCT (NCT02997202) in newly diagnosed AML; crenolanib in combination with salvage chemotherapy in R/R *FLT3*-mutated AML (NCT02400281, NCT02298166, NCT03250338), and crenolanib compared with midostaurin in newly diagnosed *FLT3*-mutated AML when used in conjunction with conventional chemotherapy (NCT03258931); quizartinib in combination with induction chemotherapy in newly diagnosed *FLT3*-mutated AML (NCT02834390, NCT03723681, NCT02668653, NCT04107727).

A major limitation in the treatment of *FLT3*-mutated AML by FLT3 inhibitor monotherapy is leukaemia relapse that often occurs within months after initial remission. In most circumstances, this is related to development of drug resistance. The mechanisms are heterogeneous and may involve emergence of clones that are resistant to FLT3 inhibitors being used; protection of leukaemia cells by BM microenvironment; and adaptation of leukaemia cells to survive FLT3 inhibitors. These mechanisms are reviewed in the following sections.

## 4. Clonal Evolution

### 4.1. Emergence of FL3-TKD Mutations

Initial evidence of emerging *FLT3*-TKD mutations as a cause of drug resistance to FLT3 inhibitors arose from laboratory studies. In particular, *FLT3*-TKD mutant clones could be selected from *in vitro* saturation mutagenesis assay [30] and found in *FLT3-*ITD AML cell lines treated with increasing dose of FLT3 inhibitors for 6–7 weeks [31]. In clinical practice, a recurrent phenomenon in patients receiving FLT3 inhibitors is the emergence of leukaemia clones carrying *FLT3*-TKD mutations at relapse. On one hand it has confirmed the on-target effect of FLT3 inhibitors and their anti-leukaemia efficacy, on the other hand it contributes to drug-resistant leukaemia relapse. Mutations may occur at the activation loop (e.g., D835, I836) or gate-keeper site (e.g., F691), and the frequencies and mutation sites depend on the specific FLT3 inhibitors being used (Table 2). Heterogeneity of *FLT3* mutant clones and polyclonal architecture with respect to *FLT3*-ITD and *FLT3*-TKD have been shown by single-cell sequencing of AML samples from patients who relapsed during quizartinib treatment [32]. In most cases *FLT3*-TKD mutations were not detectable by molecular means prior to FLT3 inhibitor treatment [10,32]. However, when pre-treatment samples were xenografted into immunodeficient mice, TKD mutant clones could emerge upon engrafting, suggesting that they might exist before treatment and were selected upon continuous exposure to inhibitor ineffective against them [10]. Nevertheless, TKD mutations only occur in 3%–60% of patients who relapsed during or after treatment with FLT3 inhibitors, and the rates are much lower in the new generation of FLT3 inhibitors e.g., crenolanib [33] and gilteritinib [34]. Moreover, in patients who relapsed after quizartinib monotherapy, single-cell analyses showed that TKD mutations occurred in up to 50% of leukaemia cells in individual patients, suggesting that they might not account entirely for the relapse [32]. Furthermore, relapses do occur upon treatment with new generation FLT3 inhibitors (e.g., gilteritinib and crenolanib) that showed inhibitory effects on TKD-mutant FLT3 proteins. Therefore, TKD mutations could only partially explain leukaemia relapse after FLT3 inhibitors. Non-TKD mediated resistance is discussed as followed.

### 4.2. Emergence of Non-FLT3 Mutations

Non-*FLT3* mutant clones have been shown to expand or emerge at relapse in *FLT3*-ITD AML during FLT3 inhibitor treatment. Next-generation sequencing of paired samples (drug-naïve sensitive and relapse drug-resistant samples) from R/R *FLT3*-ITD AML patients who relapsed from FLT3 inhibitors crenolanib [33] or gilteritinib [34] demonstrated the emergence or expansion of leukaemia clones either as subclones of the *FLT3*-ITD clone or new wildtype *FLT3* clones. These clones carried mutations of *TP53*, RAS pathway (*NRAS*, *KRAS*, *BRAF*, *PTPN11*, *CBL*), *IDH1/2, ASXL1* or *TET2*. The emergence of these mutations, particularly when they occurred in wildtype *FLT3* clones, demonstrated FLT3-independent leukaemia cells that were selected under the pressure of FLT3 inhibitors to which they were resistant. Complete loss of *FLT3*-ITD clone has been reported in up to 30% of cases [33,34].

## 5. Adaptive Cellular Mechanisms

In addition to the emergence of resistant clones, *FLT3*-ITD AML cells may adapt to FLT3 inhibitors and develop cellular mechanisms that circumvent FLT3 blockade. They include upregulation of FLT3 ligand, change in intracellular acidity and upregulation of other protein kinases. These are discussed as follows:

### 5.1. Upregulation of FLT3 Ligand

Despite the ligand independence of FLT3-ITD signalling, FLT3L has been shown to confer resistance to FLT3 inhibitors in AML [40]. FLT3L exists in soluble and membrane-bound form, the latter being expressed on stromal cells. Serum level of FLT3L was shown to increase after chemotherapy induction [40] and FLT3 inhibitor treatment [23]. FLT3L-mediated resistance to FLT3 inhibitors might be mediated through its binding to FLT3-WT receptor and activation of downstream MAPK pathway [41].

### 5.2. Increase in Intracellular pH

Microarray analysis of paired *FLT3*-ITD AML samples before sorafenib treatment and at subsequent relapse showed up-regulation of a gene encoding tescalcin (*TESC*) [42]. TESC plays a pivotal role in the maturation of sodium/hydrogen exchanger type 1 (NHE1). The latter, when activated, extrudes H+ in exchange for Na+ intake. The resulting intracellular alkalinisation provides proliferative and survival benefits to the blasts, and confers resistance by decreasing intracellular sorafenib concentration via acid–base partitioning.

### 5.3. Upregulation of other Cooperative Kinases

Upregulation of other oncogenic kinases has been shown in primary *FLT3*-ITD AML upon development of resistance to FLT3 inhibitors and may play a pathogenetic role. For instance, expression of PIM (proviral integration site for moloney murine leukaemia virus) kinase, a target of *FLT3*-ITD signalling, has been shown to increase in primary AML cells at resistance to sorafenib [43]. Overexpression of *PIM-2* in MOLM-14 (a *FLT3*-ITD AML cell line) and *FLT3*-ITD knock-in mouse model has been shown to confer resistance to quizartinib [43]. Also, expression and phosphorylation of AXL receptor tyrosine kinase was increased in PKC412(midostaurin)-resistant primary AML blasts and AML cell line [44] and its upregulation has been implicated in stroma-mediated resistance to quizartinib [45]. Pharmacological inhibition and knockout of AXL have been shown to restore sensitivity to FLT3 inhibitors-resistant AML cell lines [44]. Importantly, AXL is a therapeutic target of gilterinib [46]. Furthermore, the persistence or even up-regulation of PI3K/AKT/mTOR and MAPK/ERK pathways were shown in both FLT3 inhibitors-resistant *FLT3*-ITD AML cell lines and primary AML blasts despite inhibition of FLT3 phosphorylation by FLT3 inhibitors, suggesting that at resistance, some AML cells become independent of FLT3 signalling [47,48]. The use of inhibitors to target these compensatory pathways in combination with FLT3 inhibitors has been proposed [48] and the clinical benefits remain to be evaluated.

## 6. Microenvironment Protection

BM niche has been shown to nurture normal haematopoietic stem cell and progenitor cells and maintain steady state haematopoiesis. Mechanistic studies in mouse marrow demonstrated close interaction between osteoblasts and haematopoietic stem and progenitor cells (HSPC), suggesting a physical niche that provides signals to guide HSPC cell-fate decision [49]. Similar niche for AML may exist [50,51] and protect leukaemia cells from the inhibitory effects of FLT3 inhibitors. A number of mechanisms have been proposed (Figure 2).

### 6.1. Cytokines

BM stromal cells have been shown to secrete a repertoire of cytokines that protect the blasts from the cytotoxic effect of chemotherapy as well as FLT3 inhibitors. In addition to the FLT3L aforementioned, GM-CSF and IL-3 have been shown to protect primary *FLT3*-ITD AML cells and MV4-11 (a *FLT3*-ITD AML cell line) from crenolanib through activation of STAT5 pathway [52]. BM stromal cells also secrete fibroblast growth factor 2 (FGF2) that binds to FGFR1 in *FLT3*-ITD AML cells and promotes resistance to FLT3 inhibitors via activation of downstream MAPK pathway [53,54]. FGF2 might also bind to FGFR1 on stromal cells in an autocrine and paracrine fashion to stimulate more FGF2 secretion and stromal growth. Intriguingly, quizartinib has been shown to induce FGF2 expression in stromal cells of *FLT3*-ITD AML patients [53,54] and whether this may account for the subsequent clinical resistance to FLT3 inhibitor would have to be further investigated. Stromal-cell derived factor 1 (SDF-1) and CXCR4 axis plays a pivotal role in BM homing of HSPC. *FLT3*-ITD AML has been shown to highly express CXCR4 [55], suggesting the SDF-1/CXCR4 axis may play a pathogenetic role in this AML subtype. SDF-1 antagonist has been shown to sensitise *FLT3*-ITD transduced Ba/F3 cells (a murine pro-B lymphoid cell line) to the inhibitory effects of sorafenib when co-cultured with protective MS5 stromal cell line [56]. Activation of p53 pathway in stromal cells has been shown to reduce SDF-1 expression and abrogates the protective effect of stromal cells [57], providing a potential target for therapeutic intervention.

### 6.2. CYP3A4 in BM Stromal Cells

Cytochrome P450 3A4 (CYP3A4) is important for the metabolism and elimination of drugs and xenobiotics in human body. It is expressed primarily in hepatocytes and BM stromal cells and for the latter it might inactivate tyrosine kinase inhibitors [58] in the BM milieu. Knock-down of CYP3A4 in stromal cells or pharmacologic inhibition of CYP3A4 has been shown to ameliorate the protective effects of stromal cells both in co-culture system and in xenotransplantation model [59].

## 7. Clinical Strategies to Overcome Drug Resistance in *FLT3*-ITD AML

The mechanisms aforementioned provide the theoretical bases on which clinical trials can be designed to overcome drug resistance to FLT3 inhibitors. However, the precise mechanisms of drug resistance are likely to vary between patients and multiple mechanisms may co-exist, adding to the complexity of trial design. In clinical practice, a number of strategies have been developed to improve the treatment outcome of *FLT3*-ITD patients.

### 7.1. Upfront FLT3 Inhibitors in Combination with Chemotherapy 

Until recently, induction chemotherapy, the “7+3” regimen, has been the standard of care for fit patients of all AML subtypes including *FLT3*-ITD AML. In a Phase III randomised placebo-controlled study, addition of midostaurin to induction and consolidation chemotherapy, followed by maintenance treatment in *FLT3*-mutated AML was shown to improve overall survival [2]. The results led to FDA approval of midostaurin for the upfront treatment of *FLT3*-mutated AML. Benefits of sorafenib when added to conventional chemotherapy have also been demonstrated [60,61]. Clinical trials incorporating quizartinib (NCT02668653), crenolanib [62] and gilteritinib (NCT02236013, NCT02310321) to standard chemotherapy are underway.

CPX-351, a liposomal preparation of cytarabine and daunorubicin at a fixed synergistic drug ratio of 5:1, was recently approved as induction therapy in AML with MDS-related changes and therapy-related AML. Patients with this AML subtype receiving CPX-351 showed a superior outcome than those receiving conventional “7+3” [63]. *FLT3*-ITD AML cells have been shown to be sensitive to this preparation *in vitro* [64]. The clinical benefits of FLT3 inhibitors in combination with CPX-351 in AML with MRC or tAML carrying *FLT3*-ITD remain to be investigated.

### 7.2. Combination of FLT3 Inhibitors with Low Intensity Regimen

*FLT3*-ITD AML cells have been shown to exhibit high protein synthesis rate to maintain intracellular level of short-lived oncogenic proteins [65]. The addiction to protein synthesis provided a target for therapeutic intervention. A high-throughput *ex vivo* drug screening using primary AML cells has identified omacetaxine mepesuccinate (OME) as an effective adjunct to FLT3 inhibitors in the treatment of *FLT3*-ITD AML [65]. OME competes with t-RNA to bind to acceptor (A-site) of eukaryotic ribosome, thereby inhibiting the elongation process of protein synthesis [66]. It suppresses FLT3 downstream signalling via inhibition of the synthesis of FLT3, a short-lived protein. OME showed very acceptable toxicity profile even in the elderly but its monotherapy or combination with cytarabine showed only modest effects on AML generally. However, in combination with sorafenib (SOME), it induced remission (CR/CRi) in 72% of patients with R/R *FLT3*-ITD AML, with a deeper molecular response and extended response duration (median overall survival and leukaemia-free survival being 43.6 and 22.4 weeks respectively) among responders [20]. A clinical trial evaluating its combination with a more specific and potent FLT3 inhibitor quizartinib (QUIZOM) is in progress (NCT03135054). Preliminary observations showed that it might confer superior response and better survival, and bridge more patients to HSCT compared with SOME [67].

Furthermore, combination of FLT3 inhibitors including sorafenib, quizartinib and gilteritinib with other low intensity treatment including azacytidine or decitabine, the hypomethylating agents, have been shown to be synergistic in laboratories [68,69,70] and appeared to be effective in Phase II clinical trials [19,71,72].

### 7.3. HSCT with FLT3 Inhibitor as Maintenance Therapy

HSCT is the mainstay of treatment for *FLT3*-ITD AML in complete remission after induction chemotherapy or salvage treatments including FLT3 inhibitors [73,74,75,76,77]. However, post-HSCT relapse remains a major cause of treatment failure and may occur in up to 75% of patients. Results from clinical trials supported the proposition that maintenance with FLT3 inhibitors post-HSCT could reduce relapse and improve overall survival. In SORMAIN study, sorafenib maintenance significantly prolonged relapse-free survival [16]. Laboratory study showed that sorafenib in the post HSCT setting might increase serum IL-15 from residual *FLT3*-ITD cells that may enhance activities of allogeneic T-cells and graft-versus leukaemia effect [78]. In RADIUS Trial, patients who received midostaurin maintenance post HSCT and achieved significant FLT3 inhibition (<70% of baseline pFLT3) showed significant improvement in relapse-free and overall survivals compared with those who achieved <30% inhibition or those who received standard of care [79]. Post-HSCT maintenance with quizartinib in *FLT3*-ITD AML in a phase I study also showed reduced relapse rate [80]. Whether the benefits of FLT3 inhibitor post HSCT are related to its suppressive effects on residual *FLT3*-ITD AML cells or potentiation on graft-versus-leukaemia effects would have to be further evaluated [81].

## 8. Conclusions

Despite the availability of effective FLT3 inhibitors in the treatment of *FLT3*-ITD AML, leukaemia relapse remains to be a major cause of treatment failure. Incorporation of FLT3 inhibitor to upfront induction and consolidation chemotherapy; combination of FLT3 inhibitor with low intensity regimen as salvage treatment and the use of FLT3 inhibitor as post-HSCT maintenance may improve treatment outcome of this AML subtype. Based on findings of laboratory studies, multiple mechanisms of drug resistance have been proposed and it is likely heterogeneous among individual patients. They have provided the important bases for development of clinical trials and might raise the potential need for treatment personalisation.

## Figures and Tables

**Figure 1 ijms-21-01537-f001:**
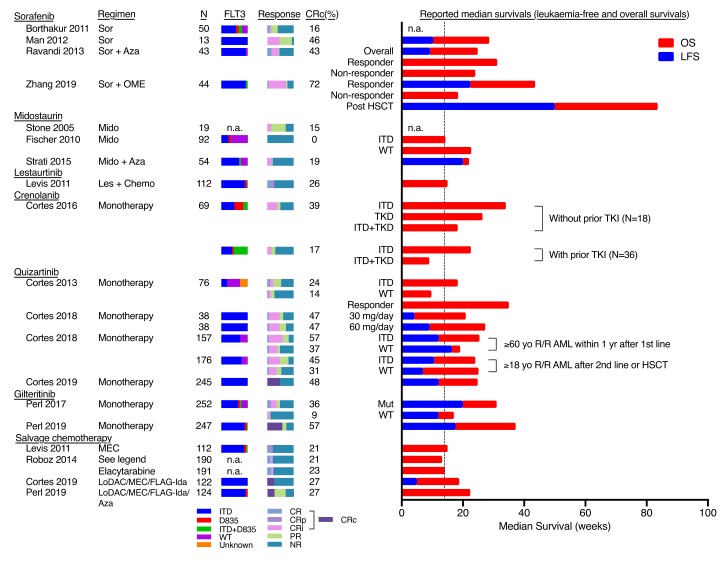
Results of clinical trials involving Fms-Like Tyrosine kinase 3 (FLT3) inhibitors on relapsed/refractory AML. The reported sample size (N), percentage of *FLT3*-ITD, D835, ITD/D835 and wildtype (WT), and response rate and duration of reported clinical trials on the use of FLT3 inhibitors on relapsed/refractory (R/R) AML were shown here. The response duration was represented as bar charts (right) for graphical presentation and it was not intended for direct statistical comparison between studies. The vertical dotted line represented the estimated pooled median overall survival in R/R patients treated with salvage chemotherapy (around 14 weeks). CR: complete remission; CRp: complete remission with incomplete platelet count; CRi: complete remission with incomplete haematological recovery; PR: partial response; NR: no response; CRc: composite complete remission rate = CR + CRp + CRi; Sor: sorafenib; Aza: Azacytidine; OME: omacetaxine mepesuccinate; Mido: midostaurin; Les: Lestaurtinib; MEC: mitoxantrone, etoposide, cytarabine; FLAG-Ida: Fludarabine, cytarabine, G-CSF, Idarubicin; LoDAC: low-dose cytarabine; HiDAC: high-dose cytarabine; TKI: tyrosine kinase inhibitors; TKI: tyrosine kinase inhibitor(s). The chemotherapy reported in Roboz et al. (2014) included investigators’ choice among 7 salvage regimens: HiDAC, MEC, FLAG/FLAG-Ida, LoDAC, hypomethylating agents, hydroxyurea, or supportive care. Reference (top to bottom): [18,10,19,20,21,17,22,23,24,25,26,27,4,28,3,23,29,4,3].

**Figure 2 ijms-21-01537-f002:**
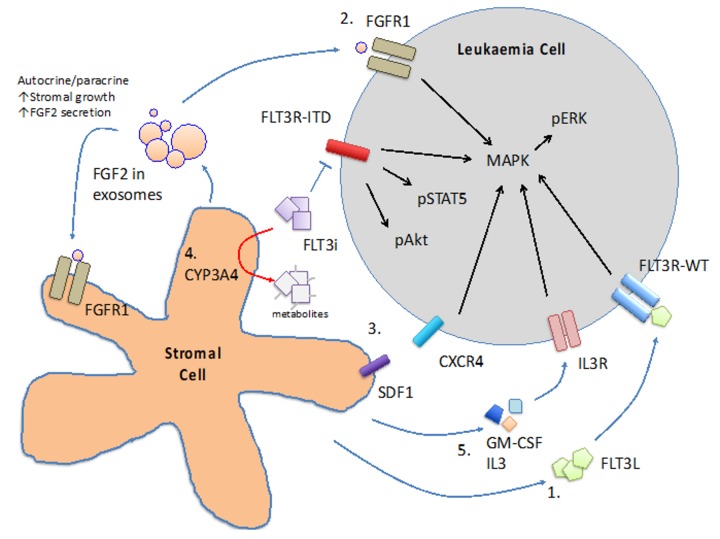
Mechanisms involved in microenvironment-mediated resistance to FLT3 inhibitors. 1. FLT3L/FLT3R-WT 2. FGF2/FGFR1 axis 3. SDF1/CXCR4 axis 4. CYP3A4-mediated degradation of FLT3 inhibitors (FLT3i) 5. GM-CSF or IL3/IL3R axis.

**Table 1 ijms-21-01537-t001:** Eight FDA-approved therapeutic agents in acute myeloid leukaemia (AML).

Therapeutic Agents	Indications
*FLT3 inhibitors*	
1. Midostaurin	Newly diagnosed *FLT3-*mutated AML
2. Gilteritinib	Relapsed/refractory *FLT3-*mutated AML
*IDH inhibitors*	
3. Ivosidenib	Relapsed/refractory *IDH1*-mutated AML
4. Enasidenib	Newly diagnosed or relapsed/refractory *IDH2*-mutated AML
*BCL2 inhibitor*	
5. Venetoclax + hypomethylating agents or LoDAC	Newly diagnosed AML aged ≥ 75
*Hedgehog pathway inhibitor*	
6. Glasdegib + LoDAC	Newly diagnosed AML aged ≥ 75
*Liposomal combination of daunorubicin and cytarabine *	
7. CPX-351	Newly diagnosed AML-MRC and t-AML
*Antibody-chemotherapy adjunct*	
8. Gemtuzumab ozogamicin	Newly diagnosed and relapsed/refractory CD33-positive AML

MRC: MDS-related changes; t-AML: transformed AML; LoDAC: low-dose cytarabine.

**Table 2 ijms-21-01537-t002:** *FLT3*-TKD mutations conferring clinical resistance to FLT3 inhibitors.

TKD Mutations	FLT3 Inhibitor Resistance
N676K	Resistance to midostaurin [35]
D835 I836 Y842	Resistance to type II FLT3 inhibitors [36,37] (sorafenib, quizartinib, ponatinib)
F691L	Resistance to crenolanib, but not to ponatinib [38] and pexidartinib [39]
K429E	Resistance to crenolanib [33]

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
