# Peer review of "Overcoming Resistance to FLT3 Inhibitors in the Treatment of FLT3-Mutated AML"

_ijms, 2020, doi:10.3390/ijms21041537_

Round 1

Reviewer 1 Report

The manuscript by Lam and Leung describes the benefits and limitations of FLT3 inhibitors in the treatment of AML patients, providing a comprehensive review of the mechanisms involved in drug resistance and/or relapse. The manuscript is well written and clearly structured. Noteworthy, the authors provide a well-conceived figure summarizing the clinical trials involving FLT3 inhibitors, that shows, for each clinical trial, all the clinical and biological information in a visually direct way.

I would only suggest to extend the section “Upregulation of other cooperative kinases” to include the different signaling pathways, frequently deregulated in AML, that are networked with FLT3 signaling, for instance PI3K/Akt and ERK among others.

Reviewer 2 Report

This is a well written and informative review on resistance to FLT3 Inhibitors in AML. I only have a few comments.

In line 53/54 the authors write that the prognostic impact of a FLT3-TKD remains unclear – which is correct – however, it is not that rare and there is a lot of data on this, the reason may rather be that its biological impact is really not that strong; anyway, please put a reference here.

Figure 1 looks nice, but in the end it is not too informative, since most oft he studies are not comparabel; however, the stacked bars imply comparability, this should be briefly mentioned/discussed

Liposomal AraC/Dauno Induction could be mentioned and its possible effect on FLT3mut AML including possible combinations in the future. 
